# Genetic Etiology of Permanent Congenital Hypothyroidism in Korean Patients: A Whole-Exome Sequencing Study

**DOI:** 10.3390/ijms26094465

**Published:** 2025-05-07

**Authors:** Jungmin Ahn, Hwalrim Jeong

**Affiliations:** 1Department of Pediatrics, School of Medicine, Jeju National University, Jeju City 63241, Republic of Korea; anny0805@hanmail.net; 2Department of Pediatrics, School of Medicine, Soonchunhyang University, Cheonan 31151, Republic of Korea

**Keywords:** congenital hypothyroidism, whole-exome sequencing, dyshormonogenesis, thyroid dysgenesis

## Abstract

Congenital hypothyroidism (CH) is among the most common endocrine disorders in neonates. Genetic testing is essential for elucidating the underlying etiology, especially in cases of permanent CH. We enrolled 32 patients diagnosed with permanent CH from the Pediatric Endocrinology Clinics at Jeju National University Hospital and Soonchunhyang University Cheonan Hospital. Whole-exome sequencing (WES) was performed on genomic DNA extracted from buccal swabs. Variants were classified according to guidelines established by the American College of Medical Genetics and Genomics (ACMG) and the Association for Molecular Pathology (AMP). WES identified 21 distinct genetic variants in 20 of the 32 patients (62.5%), spanning 6 CH-related genes: *DUOX2*, *DUOXA2*, *TPO*, *PAX8*, *TG*, and *TSHR*. Of these, 12 variants classified as pathogenic or likely pathogenic were detected in 15 patients (50%). When classified by inheritance patterns, nine patients had either homozygous (n = 1) or compound heterozygous (n = 8) variants, four patients exhibited oligogenic variants, and two patients carried a single heterozygous variant with pathogenicity. The most frequently affected gene was *DUOX2*, with pathogenic or likely pathogenic variants found in six patients. Notably, none of the six patients with thyroid agenesis or ectopic thyroid glands harbored detectable pathogenic variants. Our findings underscore the critical role of genetic analysis in determining the etiology of permanent CH. Whole-exome sequencing demonstrated a high prevalence of pathogenic variants, particularly in *DUOX2,* in Korean patients with CH. These data enhance our understanding of the genetic architecture of CH and have important implications for personalized treatment and genetic counseling.

## 1. Introduction

Primary congenital hypothyroidism (CH) is the most common endocrine disorder affecting newborns, with an estimated global incidence ranging from 1 in 1000 to 1 in 3000 live births [1]. Over the past two decades, the incidence of primary CH has nearly doubled, partly due to the implementation of improved neonatal screening programs and increased recognition of mild and transient cases, particularly those with a gland in situ (GIS). Currently, GIS accounts for approximately 35–40% of all primary CH cases [2,3]. The majority of GIS cases result from dyshormonogenesis, characterized by defects in thyroid hormone synthesis within a structurally normal thyroid gland. Dyshormonogenesis-associated congenital hypothyroidism is primarily caused by autosomal recessive mutations in genes, including *TG*, *TPO*, *DUOX2*, *DUOXA2*, *SLC26A4* (Pendred), *SLC5A5* (NIS), *IYD*, and *SLC26A7*,whereas monoallelic variants in *DUOX2* and *DUOXA2* have been frequently linked to transient CH (TCH) [4].

In contrast, thyroid dysgenesis (TD), including developmental abnormalities such as thyroid agenesis, hypoplasia, or ectopia, accounts for approximately 60–65% of primary CH cases. Although most TD cases are sporadic, genetic etiologies involving transcription factors critical for thyroid development [5]—such as *TTF1/NKX2.1*, *PAX8*, *FOXE1*, *NKX2-5*, and *GLIS3*—along with mutations in the thyroid-stimulating hormone receptor gene (*TSHR),* have been identified in approximately 5% of patients [6,7].

Genetic testing approaches, including comparative genomic hybridization (CGH) arrays, targeted next-generation sequencing (NGS), and whole-exome sequencing (WES), are increasingly utilized to elucidate the genetic basis of CH. However, studies specifically addressing the genetic heterogeneity of CH in the Korean population remain limited. Understanding the genetic factors underlying CH in diverse populations is critical for improving diagnostic accuracy, individualizing treatment, and optimizing genetic counseling.

The aim of this study was to investigate the genetic etiology of permanent CH in Korean patients using WES. By identifying pathogenic variants, we sought to enhance genetic diagnostic accuracy, clarify underlying pathophysiological mechanisms, and inform personalized management and genetic counseling strategies.

## 2. Results

### 2.1. Clinical and Biochemical Findings

A total of 32 patients with permanent congenital hypothyroidism (PCH) were included in this study. Clinical characteristics of these patients are summarized in Table 1. Thyroid ultrasonography revealed a eutopic and normal-sized thyroid gland (referred to as gland in situ, GIS) in 25 patients (78%). The remaining seven patients (22%) were diagnosed with thyroid agenesis, including thyroid agenesis (n = 3), ectopic thyroid gland (n = 3), and thyroid hypoplasia (n = 1).

At diagnosis, the median serum TSH level was 62.40 µU/mL, and the median free T4 (fT4) level was 0.84 ng/dL, consistent with congenital hypothyroidism. The median initial dose of levothyroxine (LT4) therapy was 12.68 µg/kg/day, which was subsequently adjusted according to clinical response and biochemical monitoring. Following LT4 withdrawal after 36 months of treatment, the median TSH level increased to 14.73 µU/mL, confirming permanent congenital hypothyroidism. At this stage, the median maintenance LT4 dose required to maintain euthyroidism was 44 µg/day.

### 2.2. Genetic Findings

Whole-exome sequencing identified 21 genetic variants in 20 of the 32 patients (62.5%), spanning 6 genes: *DUOX2, TPO, DUOXA2, TSHR, PAX8,* and *TG* (Table 2). Of these, 12 variants classified as pathogenic or likely pathogenic according to ACMG criteria and were identified in 15 patients (46.9%), emphasizing the clinical utility of genetic testing in this study. Allele frequencies and ACMG classification evidence codes for each variant are provided in Appendix A to facilitate transparency and reproducibility of variant interpretation.

The most frequently affected gene was *DUOX2*, with 11 distinct variants identified in 10 patients (31.3%), reinforcing its central role in dyshormonogenesis-associated congenital hypothyroidism in our study. Of these, six variants were classified as pathogenic or likely pathogenic. Notably, the p.Gly488Arg variant, classified as pathogenic, was identified in three unrelated patients (#2, #4, and #8), suggesting a potential recurrent or hotspot mutation. In addition, the p.Arg885Leu variant, classified as likely pathogenic, was detected in two patients (#4 and #7).

*DUOXA2* variants were identified in four patients (12.5%), all carrying one of two pathogenic nonsense variants, p.Tyr138Ter and p.Tyr246Ter, both of which were recurrent. Patient #12 was homozygous for the p.Tyr138Ter variant. At diagnosis, the patient exhibited markedly elevated TSH levels (>100 μIU/mL) and severely reduced fT4 (0.22 ng/dL). Although thyroid ultrasonography and scintigraphy revealed no structural abnormalities, the patient showed mild developmental delay at the time of reevaluation around 36 months of age (TSH 15.84 μIU/mL, fT4 0.75 ng/dL), which has since resolved with ongoing treatment. Patient #11, who carried the same p.Tyr138Ter variant in a heterozygous state, also presented with TSH l > 100 μIU/mL and fT4 0.30 ng/dL at diagnosis. Thyroid imaging was unremarkable. At reevaluation after discontinuation of LT4 at 36 months, TSH remained elevated (9.15 μIU/mL) with a normal fT4 level (1.25 ng/dL), and the patient has continued on LT4 therapy due to persistently elevated TSH levels.

Patients #9 and #10 were dizygotic twins, both carried the same heterozygous *DUOX2* variant (p.Leu1160del). Patient #9 additionally harbored a second *DUOX2* variant (p.Arg683Leu) and a heterozygous *TPO* variant (p.Ala863Thr), forming a potential oligogenic combination. Patient #10, in contrast, carried a heterozygous *DUOXA2* nonsense variant (p.Tyr246Ter), which was also identified in patient #13. All three patients (#9,#10, and #13) presented with similar clinical phenotypes: markedly elevated TSH levels (>100 μIU/mL), low fT4 at diagnosis, structurally normal thyroid imaging, and recurrence of hypothyroidism following LT4 withdrawal, requiring reinstitution of therapy. These clinical similarities, despite differing genotypes, suggest a potential cumulative effect of multiple variants and underscore the phenotypic convergence that can occur in genetically heterogeneous cases.

Variants in the *TG* gene were identified in two patients (6.3%), each harboring a distinct variant: p.Gln780Ter and p.Arg787Ter. Both were classified as likely pathogenic or pathogenic.

*TPO* variants were detected in three patients (9.4%). Among three distinct variants, only c.*63dup (p.?) was classified as likely pathogenic, the others were considered VUS, highlighting the need for further functional validation in *TPO*-associated mutations.

*TSHR* variants were found in three patients (9.4%). The recurrent p.Arg450His variant, classified as pathogenic, was found in two individuals (#17 and #19), both of whom had normal thyroid ultrasonography and scintigraphy findings. An additional variant, p.Ala471Thr, was identified in patient #18 and classified as a VUS. Notably, this patient has continued levothyroxine therapy (0.8 µg/kg/day) into adulthood, suggesting persistent hypothyroidism despite the uncertain pathogenicity of the variant.

*PAX8* variants were identified in two patients (6.3%), both of whom carried the same splice site variant, c.898+1G>C, classified as VUS. Patient #20 was categorized as having thyroid hypoplasia, based on ultrasound findings of a subtly small thyroid gland, while patient #19 had normal thyroid imaging and concurrently carried the pathogenic *TSHR* variant (p.Arg450His).The clinical relevance of the *PAX8* variant remains unclear and warrants further investigation.

The monogenic group consisted of 16 patients, while oligogenic variants were identified in 4 patients (#8, #9, #10, and #19), each carrying variants in two or more genes. Among these, three patients harbored at least one pathogenic or likely pathogenic variant, whereas patient #9 carried only VUSs, including compound heterozygous VUSs in *DUOX2* and an additional VUS in *TPO*. These findings suggest a possible multi-gene contribution to the phenotype, although the clinical impact of each individual variant remains uncertain.

Importantly, none of the six patients with thyroid agenesis or ectopic thyroid gland harbored pathogenic or likely pathogenic variants, supporting the notion that thyroid dysgenesis may involve non-genetic or unidentified genetic factors. Additionally, among the study subjects, three patients were born prematurely: patients #9 and #10, who were dizygotic twins born at 27 weeks of gestation, and patient #2, born at 35 weeks. Despite their prematurity, all three exhibited normal growth and neurodevelopmental outcomes. Four patients were classified as small for gestational age (SGA) at birth. Among them, two patients—one WES-positive (patient #3) and one WES-negative—exhibited persistent short stature below the third percentile and initiated growth hormone therapy. The remaining two SGA patients, including patient #6, demonstrated catch-up growth, maintaining heights around the 50th percentile. All four SGA patients also showed normal neurodevelopmental outcomes. These findings suggest that prematurity and SGA status alone may not fully account for the clinical and genetic heterogeneity observed in congenital hypothyroidism.

Collectively, these findings underscore the genetic heterogeneity of congenital hypothyroidism, demonstrating that dyshormonogenesis frequently involves identifiable genetic mutations, whereas the genetic basis of thyroid dysgenesis often remains elusive.

## 3. Discussion

In this study, we investigated the genetic etiology of permanent congenital hypothyroidism in 32 Korean pediatric patients using whole-exome sequencing. To our knowledge, this study represents the first WES-based genetic characterization of permanent CH in Korean children, demonstrating a variant detection rate of 62.5%, which aligns with previously reported rates ranging from 40% to 65% [8,9,10]. Our findings underscore the diagnostic and prognostic utility of genetic analysis in CH, providing valuable insights that may influence clinical management, genetic counseling, and therapeutic strategies, consistent with current guidelines [7,11].

Although variants were identified in 20 patients, only three cases (9.4%) were considered positive by the testing laboratory, based on the presence of homozygous or compound heterozygous pathogenic variants. The remaining cases were classified as inconclusive due to heterozygous findings in recessive genes or variants of uncertain significance. It is important to note that “inconclusive” does not preclude the potential contribution of these variants to disease. In recessive conditions, the second pathogenic allele may reside in regions not captured by exome sequencing, such as deep intronic or regulatory regions, or may involve currently unrecognized disease-associated genes. Additionally, a splice site variant in PAX8, an autosomal dominant gene, was identified, but its pathogenicity remains uncertain.

Genetic testing is increasingly recognized as an essential diagnostic tool for CH. Recent guidelines from the European Society for Pediatric Endocrinology (2020–2021) emphasize genetic evaluation, especially after careful clinical and phenotypic assessment in patients with atypical clinical presentations or a positive family history [7]. In this study, pathogenic or likely pathogenic variants were exclusively identified among patients presenting with thyroid dyshormonogenesis. Conversely, none of the seven patients diagnosed with thyroid dysgenesis, including agenesis and ectopia, had detectable pathogenic variants. These findings reinforce previous observations that thyroid dysgenesis predominantly occurs sporadically, with genetic causes documented in only about 5% of cases [7,12].

Among patients with dyshormonogenesis, heterozygous variants were identified in several key genes critically involved in thyroid hormone biosynthesis, including *DUOX2*, *DUOXA2*, *TPO*, *TG*, and *TSHR*, all of which are known to follow autosomal recessive inheritance patterns. A heterozygous splice site variant was also detected in *PAX8*, which is typically associated with autosomal dominant inheritance.

To interpret the pathogenic significance of these genetic finding, it is essential to briefly review the biochemical pathway underlying normal thyroid hormone synthesis [13]. Iodide is actively transported into thyroid follicular cells is mediated by the sodium/iodide symporter (NIS, encoded by *SLC5A5*) [14] and subsequently into the follicular lumen by apical transporters such as pendrin (*SLC26A4*) and anoctamin-1 (*ANO1*) [15]. In the lumen, iodide is oxidized by thyroid peroxidase (TPO) in the presence of hydrogen peroxide (H₂O₂), which is produced by DUOX2 and its maturation factor DUOXA2. These enzymes are essential for effective iodide oxidation and thyroid hormone production. Loss-of-function mutations in *DUOX2* or *DUOXA2* genes disrupt H₂O₂ generation, impairing thyroid hormone synthesis and resulting in congenital hypothyroidism [16]. Biallelic pathogenic variants in *DUOX2* are typically associated with permanent CH, whereas monoallelic variants may result in milder or transient forms [17,18].

In our study, notable genetic heterogeneity was observed in a pair of dizygotic twins (patients #9 and #10). Both siblings shared an identical heterozygous *DUOX2* variant (p.Leu1160del). Although classified as a variant of uncertain significance (VUS), the recurrence of the same variant in both siblings raises the possibility of shared genetic background. However, without parental genotyping, we cannot determine whether this represents familial inheritance or a coincidental finding. Further studies are needed to clarify its clinical relevance.

Despite sharing this *DUOX2* variant and exhibiting a similar initial clinical presentation and thyroid imaging finding, these twins displayed significant genetic heterogeneity and a divergent clinical course over time. The male twin (#9) carried compound heterozygous *DUOX2* variants (c.3478_3480del, c.2048G>T, p.Arg683Leu) along with an additional heterozygous *TPO* variant (c.2587G>A, p.Ala863Thr). In contrast, his female sibling (#10) harbored the same heterozygous *DUOX2* variant, but carried an additional pathogenic heterozygous *DUOXA2* mutation (c.738C>G, p.Tyr246Ter). By 9 years of age, despite similar initial clinical presentations and imaging findings, their levothyroxine (LT4) dosage requirements differed substantially: the male twin required a significantly higher LT4 dose (2.08 mcg/kg/day) compared with his female sibling (0.82 mcg/kg/day). This discrepancy might reflect differences in mutation burden, sex-related physiological variations, or other genetic and environmental modifiers influencing clinical severity. Although the additional genetic variants detected in these twins were classified as VUS, their combined contribution to the observed clinical phenotype cannot be excluded. Definitive conclusions regarding inheritance patterns and cumulative variant effects are limited by the lack of parental genetic analysis.

We also identified a heterozygous *TSHR* variant (p.Arg450His) in two patients (patients #17 and #19). This variant has been reported as a relatively common mutation associated with congenital hypothyroidism in Japan [19]. In that study, the biallelic form of p.Arg450His was associated with moderate to severe CH, while the heterozygous form was more often linked to mild or subclinical CH and a normal-sized thyroid gland. A population-based genetic epidemiological study demonstrated that this variant exhibited a founder effect in the Japanese population, accounting for approximately 67% of all detected *TSHR* mutations. Functional analyses revealed that p.Arg450His leads to reduced TSH-binding affinity and decreased cAMP-generating activity [20]. Our identification of this variant in two Korean patients (#17 and #19) underscores its potential significance within the Northeast Asian population. Given the close genetic background between Korean and Japanese populations, further population-specific studies are warranted to better elucidate its clinical relevance and prevalence in Korean patients.

In addition, we identified a novel heterozygous splice site variant in the *PAX8* gene (c.898+1G>C, p.?) in two patients (patients #19 and #20). Patient #20 presented with elevated TSH (38.73 µIU/mL) at 27 days, prompting initiation of levothyroxine treatment. At the age of 36 months, reassessment with ultrasonography and thyroid scintigraphy revealed left thyroid aplasia, indicative of thyroid dysgenesis. Defective mutations in the *PAX8* gene are generally associated with thyroid dysgenesis, most commonly presenting as thyroid hypoplasia or an ectopic thyroid gland [21,22]. The identified novel splice site mutation (c.898+1G>C) affects a critical donor splice site, likely resulting in impaired PAX8 protein function due to aberrant mRNA splicing. This patient exhibited persistent hypothyroidism and notable growth retardation (height below the third percentile) after three years of age, although developmental milestones remained within the normal range. In contrast, patient #8, who also carried this novel *PAX8* splice site variant, harbored an additional pathogenic *TSHR* variant (R450H), and showed a eutopic structurally normal thyroid gland. This clinical variability underscores the potential for incomplete penetrance or the modifying effects of co-existing genetic variants, emphasizing the need for further genotype/phenotype correlation studies.

Additionally, a novel variant in the 3′-untranslated region (3′UTR) of TPO gene (c.*63dup) in a male patient (patient #15) with dyshormonogenesis and concomitant thyroid-binding globulin deficiency. The patient presented with elevated TSH (15.4 µIU/mL) and decreased free T4 (0.5 ng/dL), and required initiation of levothyroxine (LT4) therapy at approximately 25 days of age. After withdrawal of LT4 therapy for 1 month at 36 months of age, his serum TSH significantly increased to 20.4 µIU/mL, confirming permanent hypothyroidism. Ultrasonography and thyroid scintigraphy revealed a eutopic normal-sized thyroid gland. Variants located in the 3′-untranslated region (3′UTR) of the *TPO* gene are known to influence gene expression by altering mRNA stability and translation efficiency; specifically, these variants can accelerate mRNA degradation, thereby reducing enzyme levels and impairing thyroid hormone synthesis [23]. As this represents the first reported clinical association of this variant, further studies are warranted to confirm its pathogenicity.

Although our study achieved a high mutation detection rate, several important limitations must be acknowledged. A substantial proportion of the detected variants were heterozygous findings in genes typically associated with autosomal recessive inheritance, making the direct pathogenic role of these variants uncertain. Furthermore, a heterozygous splice site variant in PAX8, a gene generally associated with autosomal dominant inheritance, was classified as a variant of uncertain significance (VUS), and its clinical relevance remains unclear. Family segregation analysis was not performed, and thus, the trans configuration of biallelic variants could not be confirmed. In addition, functional validation studies were not conducted, limiting the interpretation of novel or uncertain variants. These limitations underscore the need for parental testing, functional assays, and larger cohort studies in future research to better define the pathogenicity and inheritance patterns of the identified variants. Despite these limitations, our findings highlight the substantial genetic heterogeneity underlying permanent congenital hypothyroidism in Korean pediatric patients and emphasize the critical role of comprehensive genetic evaluation in understanding disease etiology and clinical variability. Continued advancements in genomic technologies, coupled with integrative functional analyses, will be essential to further elucidate the pathogenesis of congenital hypothyroidism and to facilitate the development of precision medicine approaches that improve patient outcomes.

In conclusion, our study highlights the substantial genetic heterogeneity underlying permanent CH in Korean pediatric patients, emphasizing the critical role of comprehensive genetic analysis in understanding disease etiology and clinical variability. Further research, including larger patient cohorts and functional assays, is essential to elucidate variant pathogenicity and uncover genetic and environmental factors contributing to thyroid dysgenesis. Continued advancements in genomic technologies will facilitate precision medicine approaches, ultimately improving patient outcomes in congenital hypothyroidism.

## 4. Materials and Methods

### 4.1. Study Design

This study included 32 patients diagnosed with permanent congenital hypothyroidism (CH) followed at the Pediatric Endocrinology Clinics at Jeju National University Hospital and Soonchunhyang University Cheonan Hospital. Patients were recruited based on persistently elevated thyroid-stimulating hormone (TSH) levels identified through neonatal screening. In Korea, neonatal screening is performed by various referral laboratories, and the TSH reference ranges may vary slightly between institutions. Therefore, infants were referred for further evaluation if their TSH levels exceeded the upper limit of the reference range defined by the screening laboratory. Upon referral to our tertiary center, levothyroxine (LT4) replacement therapy was initiated in patients whose serum TSH levels were > 10 uU/mL. To confirm permanent CH, patients underwent reevaluation after temporary discontinuation of LT4 therapy for at least one month following a minimum treatment duration of 36 months. Patients exhibiting elevated TSH levels (>10 uU/mL) upon reevaluation were classified as permanent CH cases and were enrolled in this study, whereas those maintaining normal thyroid function were considered transient CH and excluded from analysis. Serum TSH, free thyroxine (fT4), and total triiodothyronine (T3) levels were measured using an electrochemiluminescence immunoassay (ECLIA) on the cobas e 801 module (Roche Diagnostics GmbH, Mannheim, Germany). The lower detection limits were 0.005 μIU/mL for TSH, 0.5 pmol/L for fT4, and 0.195 ng/mL for T3. The reference ranges were 0.27–4.20 μIU/mL for TSH, 0.93–1.70 ng/dL for fT4, and 0.8–2.0 ng/mL for T3. All assays were performed using Elecsys reagents according to the manufacturer’s instructions. Clinical and biochemical data were analyzed using descriptive statistics. Continuous variables were assessed for normality using the Shapiro–Wilk test. Since the distribution of most clinical parameters was non-normal (*p* < 0.05), values were reported as median with interquartile range (Q1, Q3).

Thyroid morphology and glandular location were assessed in all participants using ultrasonography and thyroid scintigraphy. The perchlorate discharge test, which can assess defects in iodine organification, was not performed in this study because this diagnostic modality is not routinely available at our institutions. Patients were excluded if they present with chromosomal abnormalities, known syndromic conditions, or a family history of autoimmune thyroid disorders. The study protocol was reviewed and approved by the institutional review board (IRB) of Soonchunhyang University Cheonan Hospital (IRB No. 2023-04-026) and Jeju National University Hospital (IRB No. 2022-10-002). Informed consent was obtained from all participants and their parents. The protocol adhered to the Declaration of Helsinki and institutional guidelines.

### 4.2. Genetic Analysis

Whole-exome sequencing(WES) was performed on genomic DNA extracted from buccal swabs obtained using the AccuBuccal DNA preparation kit (AccuGene). DNA quality was assessed through agarose gel electrophoresis for integrity and using NanoDrop spectrophotometry (Thermo Fisher Scientific, Waltham, MA, USA) to ensure appropriate 260/280 (~1.8) and 260/230 (~2.1) ratios. DNA samples meeting quality criteria with a minimum yield of 300 ng, quantified using a Qubit Fluorometer (Thermo Fisher Scientific), underwent further processing.

Qualified genomic DNA samples were fragmented into 230–250 bp segments using a Covaris focused ultrasonicator. The exome library was constructed utilizing the NEXTFLEX Rapid DNA-Seq Kit (PerkinElmer, Singapore), and target regions were enriched using xGen Exome Research Panel v2 (Integrated DNA Technologies, Coralville, LA, USA). Paired-end sequencing (150 bp × 2) was conducted using an Illumina NovaSeq 6000 platform to achieve sufficient sequencing depth for robust variant identification. Copy number variant (CNV) analysis was not performed as part of this study.

Bioinformatics analyses were performed using a standardized pipeline provided by 3billion, a genetic testing company based in Seoul, Korea. Sequencing reads were aligned to the human reference genome (GRCh38) using BWA-MEM, with mitochondrial genome alignment based on the Cambridge Reference Sequence (rCRS). Variant calling was conducted using the Genome Analysis Toolkit (GATK). Variants were subsequently annotated, filtered, and prioritized using EVIDENCE, an automated in house variant interpretation system developed by 3billion. This platform integrates a daily updated variant database module, a customized ACMG/AMP-based classification module, and a symptom similarity scoring algorithm based on human phenotype ontology (HPO) terms. Annotation was performed using the Variant Effect Predictor (VEP), and variants with allele frequencies exceeding 5% in the Genome Aggregation Database (gnomAD v4.1) were excluded unless previously reported as pathogenic or likely pathogenic in ClinVar or the Human Gene Mutation Database (HGMD Professional). Variants were then classified into five categories—pathogenic (P), likely pathogenic(LP), variant of uncertain significance (VUS), likely benign (LB), or benign (B)—according to the ACMG and AMP guidelines, which were internally customized. In silico predictions for missense and splicing impact were obtained using REVEL, 3Cnet, and SpliceAI.

For the purpose of assessing the genetic contribution to CH, variants were first classified into two groups based on the number of genes affected: monogenic, in which variants were found in a single gene, and oligogenic, in which variants were identified in two or more distinct genes, suggesting potential polygenic contributions to the phenotype. Within each group, variants were further categorized by zygosity into homozygous (the same variant on both alleles), heterozygous (a single variant on one allele), and compound heterozygous(two different variants within the same genes). Finally, all identified variants were evaluated and classified according to ACMG guidelines into three categories of pathogenicity: pathogenic, likely pathogenic, and variant of uncertain significance (VUS). In addition to gnomAD v4.1, we cross-referenced each variant with the Korean Variant Archive 2 (KOVA2) database (https://www.kobic.re.kr/kova/ accessed on 4 May 2025). While many rare variants were absent from KOVA2, available allele frequency data from KOVA2 were included in Appendix A to complement East Asian frequencies from gnomAD and enhance population-specific interpretation.

## Figures and Tables

**Table 1 ijms-26-04465-t001:** Clinical characteristic of patients with permanent congenital hypothyroidism.

Parameter	Patients with CH (n = 32)
Sex (male/female), n	11/21
Age at diagnosis (days)	20.8 (13.00, 33.00)
Prematurity, n	3
Small for gestational age, n	4
Gland in situ (GIS), n	25
Thyroid hypoplasia, n	1
Thyroid agenesis, n	3
Ectopic thyroid, n	3
TSH at diagnosis (µIU/mL)	62.4 (17.08, 100.0)
fT4 at diagnosis (ng/dL)	0.84 (0.61, 1.13)
LT4 dose at diagnosis (µg/kg/day)	12.68 (10.13, 15.82)
TSH at reevaluation (µIU/mL) fT4 at reevaluation (ng/dL)	14.73 (10.40, 20.90) 1.26 (1.00, 1.31)
LT4 dose at reevaluation (µg/day)	44.00 (34.00, 52.50)

Data are presented as median (Q1, Q3) unless otherwise indicated. Abbreviations: CH—congenital hypothyroidism; GIS—gland in situ; fT4—free thyroxine; LT4—levothyroxine; TSH—thyroid-stimulating hormone; Q1—first quartile; Q3—third quartile.

**Table 2 ijms-26-04465-t002:** Genetic variants identified in patients with permanent congenital hypothyroidism.

Patient No.	Gene	Variants (NM Accession; cDNA/Protein)	Inheritance	Zygosity	ACMG Classification
1	*DUOX2*	NM_001363711.2:c.714C>A (p.Tyr238Ter)	AR	Heterozygous	Likely pathogenic
2	*DUOX2*	NM_001363711.2:c.1462G>A (p.Gly488Arg)	AR	Heterozygous	Pathogenic
3	*DUOX2*	NM_001363711.2:c.3721A>T (p.Ile1241Phe)	AR	Heterozygous	VUS
4	*DUOX2*	NM_001363711.2:c.1462G>A (p.Gly488Arg) NM_001363711.2:c.2654G>T (p.Arg885Leu)	AR	Compound heterozygous	Pathogenic/ Likely pathogenic
5	*DUOX2*	NM_001363711.2:c.2000del (p.Leu667ArgfsTer9)/ NM_001363711.2:c.3184+1G>A (p.?)	AR	Compound heterozygous	Pathogenic/ Likely pathogenic
6	*DUOX2*	NM_001363711.2:c.1588A>T (p.Lys530Ter)/ NM_001363711.2:c.4408C>T (p.Arg1470Trp)	AR	Compound heterozygous	Pathogenic/ VUS
7	*DUOX2*	NM_001363711.2:c.2654G>T (p.Arg885Leu)/ NM_001363711.2:c.2428G>A (p.Glu810Lys)	AR	Compound heterozygous	Likely pathogenic/ VUS
8	*DUOX2/* *TG*	NM_001363711.2:c.1462G>A (p.Gly488Arg) NM_003235.5:c.2359C>T (p.Arg787Ter)	AR/ AR	Heterozygous/ Heterozygous	Pathogenic/ Pathogenic
9	*DUOX2/* *DUOX2/* *TPO*	NM_001363711.2:c.3478_3480del (p.Leu1160del)/ NM_001363711.2:c.2048G>T (p.Arg683Leu)/ NM_001206744.2:c.2587G>A (p.Ala863Thr)	AR/ AR	Compound heterozygous/ Heterozygous	VUS/ VUS/ VUS
10	*DUOX2/* *DUOXA2*	NM_001363711.2:c.3478_3480del (p.Leu1160del) / NM_207581.4:c.738C>G (p.Tyr246Ter)	AR/ AR	Heterozygous/ Heterozygous	VUS/ Pathogenic
11	*DUOXA2*	NM_207581.4:c.413dup (p.Tyr138Ter)	AR	Heterozygous	Pathogenic
12	*DUOXA2*	NM_207581.4:c.413dup (p.Tyr138Ter)	AR	Homozygous	Pathogenic
13	*DUOXA2*	NM_207581.4:c.738C>G (p.Tyr246Ter)	AR	Heterozygous	Pathogenic
14	*TG*	NM_003235.5:c.2338C>T (p.Gln780Ter)	AR	Heterozygous	Likely pathogenic
15	*TPO*	NM_001206744.2:c.*63dup (p.?)	AR	Heterozygous	Likely pathogenic
16	*TPO*	NM_001206744.2:c.2017G>A (p.Glu673Lys)	AR	Heterozygous	VUS
17	*TSHR*	NM_000369.5:c.1349G>A (p.Arg450His)	AR	Heterozygous	Pathogenic
18	*TSHR*	NM_000369.5:c.1411G>A (p.Ala471Thr)	AR	Heterozygous	VUS
19	*TSHR/* *PAX8*	NM_000369.5:c.1349G>A (p.Arg450His)/ NM_003466.4:c.898+1G>C (p.?)	AR/ AD	Heterozygous/ Heterozygous	Pathogenic/ VUS
20	*PAX8*	NM_003466.4:c.898+1G>C (p.?)	AD	Heterozygous	VUS

Abbreviations: AR—autosomal recessive; AD—autosomal dominant; VUS—variant of uncertain significance; ACMG—American College of Medical Genetics and Genomics. Detailed ACMG criteria and allele frequencies for each variant are provided in Appendix A.

## Data Availability

The data presented in this study are not publicly available due to privacy and ethical restrictions.

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
