# Peer review of "Genetic Etiology of Permanent Congenital Hypothyroidism in Korean Patients: A Whole-Exome Sequencing Study"

_ijms, 2025, doi:10.3390/ijms26094465_

Round 1
Reviewer 1 Report
Comments and Suggestions for Authors
Ahn and Jeong performed whole exome sequencing in 32 infants with permanent congenital hypothyroidism (CH) and identified pathogenic or likely pathogenic variants in CH-related genes.
General comments:
This article highlights the importance of performing genetic testing in infants with CH and adds information to the genetic causes of CH. However, some serum parameters to describe thyroid function are missing.
The tables 1 and 2 are duplicated on pages 6-8. I suggest to present the prevalence of likely pathogenic and pathogenic variants in this collective according to the mode of inheritance (1. homozygous, compound heterozygous, 2. oligogenic, 3. heterozygous. The conclusion, especially for heterozygous or polygenic results, must be drawn more carefully.
Specific comments:
Abstract:
Page 1, line 19:
I suggest to present the prevalence of likely pathogenic and pathogenic variants in this collective according to the mode of inheritance (1. homozygous, compound heterozygous, 2. oligogenic, 3. heterozygous. These two groups should be discussed separately.
Materials and Methods:
Page 2, line 70: please provide cut-off of TSH at inclusion
Page 2, line 74: please provide fT4 value at reevaluation. Was thyreoglobuline measured? Please provide the assay which was used for measuring thyroid hormone values.
Page 3, line 116: please provide statitics for calculation mean in table 1
Results:
Page 3, line 122: 78% with normal-sized thyroid gland seems very high.
Page 3, line 125: please provide normal values for fT4
Page 3, line 131, table 1: please provide SD and range and normal values for all measurements
Page 3, line 128: please provide fT4 values
Page 4, line 132: table 2: I suggest to create two tables. One with the results in patients with homozygous or compound heterozygous variants and the other with patients with oligogenic heterozygous and monogenic heterozygous variants.
Page 6, line 144: The tables 1 and 2 are duplicated on pages 6-8.
Page 8, line 147: apart from PAX8, all genes are known for autosomal recessive. Therefore, I recommend to describe the patients with homozygous or compound heterzogous in one group and discuss this collective separately. Also the prevalence should be mentioned: 5 of 32 (15,6%) patients harbored a homozygous or compound heterozygous pathogenic or likely pathogenic variant (patient 1,2,3,5,6) all in DUOX2. I would carefully describe patient 7 and 16 with a pathogenic and VUS in DUOX2.
I recommend that you speculate more carefully that it could be a dominant inheritance or incomplete penetrance. Without analyzing the parents there is no proof. It is more likely that the detected heterozygous genes found have no biological effect on the thyroid function. This should also be discussed.
Discussion:
Page 8, line 167: as mentioned above, I suggest to state, that the detection rate was 15,6%.
Page 8, line 182: this are heterozygous variants in known autosomal recessive inheritance (apart PAX8). This should be stated.
Page 8, line 185: I fell the description of DUOX2 can be shortened
Page 8, line 200: Due to missing data of the parents it is speculation and should be interpreted more carefully.
Page 8, line 220: the cited manuscript described this TSHR variant in a biallelic mutation to cause moderate to severe CH. This should be stated
Author Response
Dear reviewers,
I have revised the manuscript as requested in the comments.
The answers are highlighted in yellow.
General comments:
This article highlights the importance of performing genetic testing in infants with CH and adds information to the genetic causes of CH. However, some serum parameters to describe thyroid function are missing.
The tables 1 and 2 are duplicated on pages 6-8. I suggest to present the prevalence of likely pathogenic and pathogenic variants in this collective according to the mode of inheritance (1. homozygous, compound heterozygous, 2. oligogenic, 3. heterozygous. The conclusion, especially for heterozygous or polygenic results, must be drawn more carefully.
è Thank you for your valuable feedback. As suggested, we revised the manuscript to clarify the classification of variants. The following sentence has been added to the Methods section.
“For the purpose of assessing the genetic contribution to CH, variants were first classified into two groups based on the number of genes affected: monogenic, in which variants were found in a single gene, and oligogenic, in which variants were identified in two or more distinct genes, suggesting a potential polygenic contributions to the phenotype. Within each group, variants were further categorized by zygosity into homozygous (the same variant on both alleles), heterozygous ( a single variant on one allele), and compound heterozygous(two different variants within the same genes). Finally, all identified variants were evaluated and classified according to ACMG guidelines into the three categorize of pathogenicity; pathogenic, likely pathogenic, and variant of uncertain significance (VUS).“
Specific comments:
Abstract:
Page 1, line 19:
I suggest to present the prevalence of likely pathogenic and pathogenic variants in this collective according to the mode of inheritance (1. homozygous, compound heterozygous, 2. oligogenic, 3. heterozygous. These two groups should be discussed separately.
è Thank you for your valuable feedback. As suggested, we revised the abstract to reflect the prevalence of pathogenic and likely pathogenic variants according to the mode of inheritance.
“Results: WES identified 21 distinct genetic variants in 20 of the 32 patients (62.5%), spanning six CH- related genes: DUOX2, DUOXA2, TPO, PAX8, TG, and TSHR. Of these, 12 variants classified as pathogenic or likely pathogenic were detected in 15 patients (50%). When classified by inheritance pattern, 9 patients had either homozygous (n=1) or compound heterozygous (n=8) variants, 4 patients exhibited oligogenic variants, and 2 patients carried a single heterozygous variant with pathogenicity. The most frequently affected gene was DUOX2, with pathogenic or likely pathogenic variants found in 6 patients. Notably, none of the six patients with thyroid agenesis or ectopic thyroid gland harbored detectable pathogenic variants.”
Materials and Methods:
Page 2, line 70: please provide cut-off of TSH at inclusion
è Thank you for your comment. We agree that the TSH cut-off is important; however, in Korea, neonatal screening for congenital hypothyroidism is performed by various referral laboratories, and the cut-off values may differ slightly depending on the institution.
Patients included in this study were referred to our tertiary centers if their TSH levels exceeded the reference range defined by the screening laboratory. Therefore, the TSH cut-off values at the time of screening were not uniform but were above the upper limit of normal established by each testing facility.
As suggested, we revised the Method section to clarify the inclusion criteria.
“Patients were recruited based on persistently elevated thyroid-stimulating hormone (TSH) levels identified through neonatal screening. In Korea, neonatal screening is performed by various referral laboratories, and the TSH reference ranges may vary slightly between institutions. Therefore, infants were referred for further evaluation if their TSH levels exceeded the upper limit of the reference range defined by the screening laboratory. Upon referral to our tertiary center, levothyroxine (LT4) replacement therapy was initiated in patients whose serum TSH levels were >10uU/mL.”
Page 2, line 74: please provide fT4 value at reevaluation. Was thyroglobuline measured? Please provide the assay which was used for measuring thyroid hormone values.
è Thank you for your thoughtful comments. The reevaluation Ft4 values are -----------/ Thyroglobulin was not measured, and therefore was not included in the current. As requested, we have added a description of the thyroid hormone assays to the Methods section as follows.
“ Serum TSH, free thyroxine (fT4), and total triiodothyronine (T3) levels were measured using an electrochemiluminescence immunoassay (ECLIA) on the cobas e 801 module (Roche Diagnostics GmbH, Mannheim, Germany). The lower detection limits were 0.005 μIU/mL for TSH, 0.5 pmol/L for fT4, and 0.195 ng/mL for T3. The reference ranges were 0.27–4.20 μIU/mL for TSH, 0.93–1.70 ng/dL for fT4, and 0.8–2.0 ng/mL for T3. All assays were performed using Elecsys reagents according to the manufacturer’s instructions.”
Page 3, line 116: please provide statitics for calculation mean in table 1
=> Thank you for your comment. We have added a description of the statistical methods used to calculate the mean, standard deviation, and range in the Methods section as follows:
“Clinical and biochemical data were analyzed using descriptive statistics. Continuous variables were assessed for normality using the Shapiro–Wilk test. Since the distribution of most clinical parameters was non-normal (p < 0.05), values are reported as median with interquartile range (Q1, Q3).”
Results:
Page 3, line 122: 78% with normal-sized thyroid gland seems very high.
è Thank you for your observation. We agree that this proportion appears relatively high. However, the majority of patients in our study were suspected to have dyshormonogenesis, which typically presents with a structurally normal and eutopic thyroid gland. This distribution is consistent with previous reports in East Asian populations, where dyshormonogenesis is more common than thyroid dysgenesis.
Page 3, line 125: please provide normal values for fT4
è Thank you for your comment. We revised the sentence as follows:
“At diagnosis, the median serum TSH level was 66 µU/mL, and the mean free T4 (fT4) level was 0.9 ng/dL (reference range: 0.93-1.7 ng/dL), findings consistent with congenital hypothyroidism.”
Page 3, line 131, table 1: please provide SD and range and normal values for all measurements
è Thank you for your comment. As requested, we revised Table 1 to include the values for fT4 and reanalyzed all measurements; since the data did not follow a normal distribution, results are now presented as median with interquartile range (Q1, Q3) instead of mean values.
Page 3, line 128: please provide fT4 values
è Thank you for your comment. As requested, we revised Table 1 to include the values for fT4 and reanalyzed all measurements; since the data did not follow a normal distribution, results are now presented as median with interquartile range (Q1, Q3) instead of mean values.
Page 4, line 132: table 2: I suggest to create two tables. One with the results in patients with homozygous or compound heterozygous variants and the other with patients with oligogenic heterozygous and monogenic heterozygous variants.
è Thank you for your helpful suggestion. Instead of creating two separate tables, we chose to present all data in a single comprehensive table, organized primarily by gene. This structure reflects the clinical and genetic relevance of each gene in CH. Within each gene category, patients are further distinguished based on the number of variants detected (e.g., single, biallelic, of oligogenic)and their zygosity, which is clearly indicated in the adjacent columns, We believe this gene-centered approach provides a clearer overview of genotype distribution and facilitates interpretation without fragmenting the data across multiple table.
Page 6, line 144: The tables 1 and 2 are duplicated on pages 6-8.
è Thank you for pointing this out. The duplication appears to have occurred during the editorial formatiing process. We have corrected this issue and ensured that each table is presented only once in the revise manuscript.
Page 8, line 147: apart from PAX8, all genes are known for autosomal recessive. Therefore, I recommend to describe the patients with homozygous or compound heterzogous in one group and discuss this collective separately. Also the prevalence should be mentioned: 5 of 32 (15,6%) patients harbored a homozygous or compound heterozygous pathogenic or likely pathogenic variant (patient 1,2,3,5,6) all in DUOX2. I would carefully describe patient 7 and 16 with a pathogenic and VUS in DUOX2.
I recommend that you speculate more carefully that it could be a dominant inheritance or incomplete penetrance. Without analyzing the parents there is no proof. It is more likely that the detected heterozygous genes found have no biological effect on the thyroid function. This should also be discussed.
è Thank you for your important and constructive feedback. We fully agree that the interpretation of heterozygous variants, particularly in genes typically associated with autosomal recessive inheritance, should be approached with caution, particularly in the absence of parental data. Accordingly, we have extensively revised the Result section as followes.
“Whole-exome sequencing identified 21 genetic variants in 20 of the 32 patients (62.5%), spanning six gene: DUOX2, TPO, DUOXA2, TSHR , PAX8, and TG (Table 2). Of these, 12 variants classified as pathogenic or likely pathogenic according to ACMG criteria and were identified in 15 patients (46.9%), underscoring the clinical utility of genetic testing in this study.
The most frequently affected gene was DUOX2, with 11 distinct variants identified in 10 patients (31.3%), reinforcing its central role in dyshormonogenesis-associated congenital hypothyroidism in our study. Of these, six variants were classified as pathogenic or likely pathogenic. Notably, the p.Gly488Arg variant, classified as pathogenic, was identified in three unrelated patients (#2, #4, and #8), suggesting a potential recurrent or hotspot mutation. In addition, the p.Arg885Leu variant, classified as likely pathogenic, was detected in two patients (#4 and #7).
DUOXA2 variants were identified in four patients (12.5%), all carrying one of two of pathogenic nonsense variants, p.Tyr138Ter and p.Tyr246Ter, both of which were recurrent. Patient #12 was homozygous for the p.Tyr138Ter variant. At diagnosis, the patient exhibited markedly elevated TSH levels (>100 μIU/mL) and severely reduced fT4 (0.22 ng/dL). Although thyroid ultrasonography and scintigraphy revealed no structural abnormalities, the patient showed mild developmental delay at the time of reevaluation around 36 months of age (TSH 15.84 μIU/mL, fT4 0.75 ng/dL), which has since resolved with ongoing treatment. Patient #11, who carried the same p.Tyr138Ter variant in a heterozygous state, also presented with TSH l>100 μIU/mL and fT4 0.30 ng/dL at diagnosis. Thyroid imaging was unremarkable. At reevaluation after discontinuation of LT4 at 36 months, TSH remained elevated (9.15 μIU/mL) with a normal fT4 level (1.25 ng/dL), and the patient has continued on LT4 therapy due to persistently elevated TSH levels.
Patients #9 and #10 were dizygotic twins, both carried the same heterozygous DUOX2 variant (p.Leu1160del). Patient #9 additionally harbored an second DUOX2 variant (p.Arg683Leu) and a heterozygous TPO variant (p.Ala863Thr), forming a potential oligogenic combination. Patient #10, in contrast, carried a heterozygous DUOXA2 nonsense variant (p.Tyr246Ter), which was also identified in patient #13. All three patients (#9,#10, and #13) presented with similar clinical phenotypes: markedly elevated TSH levels (>100 μIU/mL), low fT4 at diagnosis, structurally normal thyroid imaging, and recurrence of hypothyroidism following LT4 withdrawal, requiring reinstitution of therapy. These clinical similarities, despite differing genotypes, suggest a potential cumulative effect of multiple variants and underscore the phenotypic convergence that can occur in genetically heterogeneous cases.
Variants in the TG gene were identified in two patients (6.3%), each harboring a distinct variant: p.Gln780Ter and p.Arg787Ter. Both were classified as likely pathogenic or pathogenic.
TPO variants were detected in three patients (9.4%). Among three distinct variants, only c.*63dup (p.?) was classified as likely pathogenic, the others were considered VUS, highlighting the need for further functional validation in TPO-associated mutations.
TSHR variants were found in three patients (9.4%).The recurrent p.Arg450His variant, classified as pathogenic, was found in two individuals (#17 and #19), both of whom had normal thyroid ultrasonography and scintigraphy findings. An additional variant, p.Ala471Thr, was identified in patient #18 and classified as a VUS. Notably, this patient has continued levothyroxine therapy (0.8 µg/kg/day) into adulthood, suggesting a persistent hypothyroidism despite the uncertain pathogenicity of the variant.
PAX8 variants were identified in two patients (6.3%), both of whom carried the same splice-site variant, c.898+1G>C, classified as VUS. Patient #20 was categorized as having thyroid hypoplasia, based on ultrasound findings of a subtly small thyroid gland, while patient #19 had normal thyroid imaging and concurrently carried the pathogenic TSHR variant (p.Arg450His).The clinical relevance of the PAX8 variant remains unclear and warrants further investigation.
The monogenic group consisted of 16 patients, while oligogenic variants were identified in four patients (#8, #9, #10, and #19), each carrying variants in two or more one gene. Among these, three patients harbored at least one pathogenic or likely pathogenic variant, whereas patient #9 carried only VUSs, including compound heterozygous VUSs in DUOX2 and additional VUS in TPO. These finding suggest a possible multi-gene contribution to the phenotype, although the clinical impact of each individual variant remains uncertain.
Importantly, none of the six patients with thyroid agenesis or ectopic thyroid gland harbored pathogenic or likely pathogenic variants, supporting the notion that thyroid dysgenesis may involve non-genetic or unidentified genetic factors.
Collectively, these findings underscore the genetic heterogeneity of congenital hypothyroidism, demonstrating that dyshormonogenesis frequently involves identifiable genetic mutations, whereas the genetic basis of thyroid dysgenesis often remains elusive. “
.
Discussion:
Page 8, line 167: as mentioned above, I suggest to state, that the detection rate was 15,6%.
è Thank you for your thoughtful comment. We agree that the interpretation of genetic variants should prioritize clinical significance, particularly in autosomal recessive genes where biallelic pathogenic variants are most indicative of causality.
In our study, genetic variants were identified in 20 of the 32 patients (62.5%), which reflects the overall detection rate of variants, including pathogenic, likely pathogenic, and VUS. However, as noted, only three patients (#4, #5, and #13) were classified as having positive results by the testing laboratory (3billion), based on the presence of homozygous or compound heterozygous pathogenic variants with clear diagnostic value. The remaining 17 cases were categorized as inconclusive, often due to heterozygous findings in recessive genes or the presence of variants of uncertain significance. We acknowledge this distinction in the revised Discussion and clarified that while the variant detection rate was 62.5%, only a subset of cases met strict diagnostic criteria for pathogenicity. This distinction highlights the importance of standardized variant interpretation frameworks and reflects ongoing challenges in translating WES findings into clinical practice.
We modified the paragraph as follows :
“In this study, we investigated the genetic etiology of permanent congenital hypothyroidism in 32 Korean pediatric patients using whole-exome sequencing. To our knowledge, this study represents the first WES-based genetic characterization of permanent CH in a Korean children, demonstrating a variant detection rate of 62.5%, which aligns with previously reported rates ranging from 40% to 65% (8-10). Our findings underscore the diagnostic and prognostic utility of genetic analysis in CH, providing valuable insights that may influence clinical management, genetic counseling, and therapeutic strategies, consistent with current guidelines (7, 11).
Although variants were identified in 20 patients, only three cases (9.4%) were considered positive by the testing laboratory, based on the presence of homozygous or compound heterozygous pathogenic variants. The remaining cases were classified as inconclusive due to heterozygous findings in recessive genes or variants of uncertain significance. It is important to note that “inconclusive” does not preclude the potential contribution of these variants to disease. In recessive condition, the second pathogenic allele may reside in regions not captured by exome sequencing, such as deep intronic or regulatory regions, or may involve currently unrecognized disease-associated genes. Additionally, a splice-site variant in PAX8, an autosomal dominant gene, was identified, but its pathogenicity remains uncertain. .”
“
Page 8, line 182: this are heterozygous variants in known autosomal recessive inheritance (apart PAX8). This should be stated.
è Thank you for your suggestion. As recommended, we have revised the sentence to clarify the inheritance patterns of the genes involved.
“Among patients with dyshormonogenesis, heterozygous variants were identified in several key genes critically involved in thyroid hormone biosynthesis, including DUOX2, DUOXA2, TPO, TG, and TSHR, all of which are known to follow autosomal recessive inheritance patterns. A heterozygous splice-site variant was also detected in PAX8 , which is typically associated with autosomal dominant inheritance”
Page 8, line 185: I fell the description of DUOX2 can be shortened
è Thank you for your helpful suggestion. As recommended, we have shortened the description of DUOX2 and its role in thyroid hormone synthesis to improve clarity and focus.
“ To interpret the pathogenic significance of these genetic finding, it is essential to briefly review the biochemical pathway underlying normal thyroid hormone synthesis (13).Iodide is actively transported into thyroid follicular cells is mediated by the sodium-iodide symporter (NIS, encoded by SLC5A5)(14) and subsequently into the follicular lumen by apical transporters such as pendrin (SLC26A4) and anoctamin-1 (ANO1)(15). In the lumen, iodide is oxidized by thyroid peroxidase (TPO) in the presence of hydrogen peroxide (Hâ‚‚Oâ‚‚), which is produced by DUOX2 and its maturation factor DUOXA2. These enzyme are essential for effective iodide oxidation and thyroid hormone production. Loss-of-function mutations in DUOX2 or DUOXA2 genes disrupt Hâ‚‚Oâ‚‚ generation, impairing thyroid hormone synthesis and resulting in congenital hypothyroidism (16). Biallelic pathogenic variants in DUOX2 are typically associated with permanent CH, whereas monoallelic variants may result in milder or transient forms”
Page 8, line 200: Due to missing data of the parents it is speculation and should be interpreted more carefully.
è Thank you for your comment. We agree that interpretation of recurrence in sibilings should be made with caution in the absence of parental data. The revised sentence now reads:
“In our study, notable genetic heterogeneity was observed in a pair of dizygotic twins (Patients #9 and #10). Both siblings shared an identical heterozygous DUOX2 variant (p.Leu1160del). Although classified as a variant of uncertain significance (VUS), the recurrence of the same variant in both siblings raises the possibility of shared genetic background. However without parental genotyping, we cannot determine whether this represents familial inheritance or a coincidental finding. Further studies are needed to clarify its clinical relevance”
Page 8, line 220: the cited manuscript described this TSHR variant in a biallelic mutation to cause moderate to severe CH. This should be stated
è Thank you for your comment. As suggested, we have revised the sentence to reflect previous reports describing the TSHR p.Arg450His variant in the biallelic state as a cause of moderate to severe congenital hypothyroidism. The updated sentence now reads:
“We also identified a heterozygous TSHR variant (p.Arg450His) in two patients (Patient #17 and #19). This variant has been reported as a relatively common mutation associated with congenital hypothyroidism in Japan(19). In that study, the biallelic form of p.Arg450His was associated with moderate to severe CH, while the heterozygous form was more often linked to mild or subclinical CH and a normal-sized thyroid gland. A population-based genetic epidemiological study demonstrated that this variant exhibited a founder effect in the Japanese population, accounting for approximately 67% of all detected TSHR mutations. Functional analyses revealed that p.Arg450His leads to reduced TSH-binding affinity and decreased cAMP-generating activity (20). Our identification of this variant in two Korean patients(#17 and #19) undescore its potential significance within the Northeast Asian population. Given the close genetic background between Korean and Japanese populations, further population-specific studies are warranted to better elucidate its clinical relevance and prevalence in Korean patients.”

Reviewer 2 Report
Comments and Suggestions for Authors
The authors used whole-exome sequencing to analyze 32 Korean patients with permanent congenital hypothyroidism and identified pathogenic variants in 50% of the cases - most commonly in the DUOX2 gene. Although the study is interesting, the following recommendations should be carefully considered to improve the manuscript.
- Reads should be aligned to GRCh38 according to GATK recommendations.
- Is 3 billion a company? If so, please cite the city and country.
- Include a reference that supports the performance of the EVIDENCE system.
- Have only congenital hypothyroidism-associated genes been analyzed?
- Indicate the source of population-matched allele frequencies and the identity of in silico predictors used to assess variant pathogenicity.
- Were CNVs tested?
- Was thyroid size tested in patients with GIS?
- Was the perchlorate discharge test positive in patients with DUOX or TPO variants?
- Family segregation of variants classified as biallelic is encouraged to confirm that variants are in trans.
- Table 1: Include number of patients with GIS and range of hormone values at diagnosis and withdrawal.
- Table 2: Include criteria and strength of evidence for ACMG classification
- Tables 1 and 2 are duplicated.
- Data presentation should be sufficiently clear to allow correlation between clinical findings, biochemistry, and genetics. As the information is currently presented, such correlations cannot be made.
- As presented, data on patients with thyroid dysgenesis are misleading. Genetic testing revealed a VUS variant in Pax8 in patient #20 with thyroid dysgenesis.
Author Response
Dear reviewers,
I have revised the manuscript as requested in the comments.
The answers are highlighted in yellow.
The authors used whole-exome sequencing to analyze 32 Korean patients with permanent congenital hypothyroidism and identified pathogenic variants in 50% of the cases - most commonly in the DUOX2 gene. Although the study is interesting, the following recommendations should be carefully considered to improve the manuscript.
- Reads should be aligned to GRCh38 according to GATK recommendations.
èIt has been revised accordingly and the change has been highlighted in the Table2.
- Is 3 billion a company? If so, please cite the city and country.
è Yes, 3billion is a genetic testing company based in Seoul, Republic of Korea. The city and country have been added in the revised manuscript as follows:
” (3 billion- a genetic testing company, Seoul, Korea).”
- Include a reference that supports the performance of the EVIDENCE system.
è The Evidence system is an in-house automated variant interpretation pipeline developed by 3billion. It incorporates daily-updated variant and disease databases integrates patient phenotype data using HPO terms, and prioritizes variants based on symptom similarity scores. We have added a detailed description of this system to the Methods section, as shown below:
“Bioinformatics analyses were performed using a standardized pipeline provided by 3 billion, a genetic testing company based in Seoul, Korea. Sequencing reads were aligned to the human reference genome (GRCh37) using BWA-MEM, with mitochondrial genome alignment based on the Cambridge Reference Sequence (rCRS). Variants calling was conducted using the Genome Analysis Toolkit (GATK). Variants were subsequently annotated, filtered, and prioritized using EVIDENCE, an automated in house variant interpretation system developed by 3billoion. This platform integrates a daily-updated variant databases module, a customized AVMG/AMP-based classification module, and a symptom similarity scoring algorithm based on Human Phenotype Ontology (HPO) terms. Annotation was performed using the Variant Effect Predictor (VEP), and variants with allele frequencies exceeding 5% in the Genome Aggregation Database (gnomAD v3.1) were excluded unless previously reported as pathogenic or likely pathogenic in ClinVar or the Human Gene Mutation Database(HGMD Professional). Variants were then classified into five categories-pathogenic (P), likely pathogenic(LP), variant of uncertain significance (VUS), likely benign (LB), or benign (B), according to the ACMG and AMP guidelines, which were internally customized. In Silico predictions for missense and splicing impact were obtained using REVEL, 3Cnet, and SpliceAI”
- Have only congenital hypothyroidism-associated genes been analyzed?
è Thank you for your question. We performed whole-exome sequencing, and therefore, the analysis was not limited to congenital hypothyroidism-associated genes.
- Indicate the source of population-matched allele frequencies and the identity of in silico predictors used to assess variant pathogenicity.
è Thank you for your comment. The sources of population-matched allele frequencies and the in silico prediction tools have been specified in the revised Methods section and are now clearly indicated in the manuscript.
- Were CNVs tested?
è Thank you for your question. Copy number variant (CNV) analysis was not performed in this study.
- Was thyroid size tested in patients with GIS?
è Thank you for your question. Precise thyroid size measurements were not available for patients with GIS, as the data were obtained retrospectively from older records.
- Was the perchlorate discharge test positive in patients with DUOX or TPO variants?
è Thank you for your question. The perchlorate discharge test was not performed in patients with DUOX2 or TPO variants.
- Family segregation of variants classified as biallelic is encouraged to confirm that variants are in trans.
è Thank you for your insightful comment. Family testing was not performed in this study; therefore, segregation analysis to confirm trans configuration of biallelic variants was not available. We acknowledge the importance of this approach and appreciate your suggestion.
- Table 1: Include number of patients with GIS and range of hormone values at diagnosis and withdrawal.
è Thank you for your comment. As suggested, hormone values at diagnosis and withdrawal have been revised and are now presented as median (Q1, Q3) in Table 1. We have clarified the distinction between patients with gland-in-situ (GIS) and thyroid dysgenesis in the Results section, specifying the number of patients with thyroid agenesis, ectopic thyroid, and hypoplasia.
“Thyroid ultrasonography revealed a eutopic and normal-sized thyroid gland (referred to as gland-in situ, GIS) in 25 patients (78%). The remaining seven patients (22%) were diagnosed with thyroid agenesis, including thyroid agenesis(n=3), ectopic thyroid gland (n=3), and thyroid hypoplasia (n=1).”
- Table 2: Include criteria and strength of evidence for ACMG classification
è Thank you for your comment. In Table 2, we have provided the final ACMG classification (pathogenic, likely pathogenic, or variant of uncertain significance) for each variant, in accordance with the ACMG/AMP guidelines. While individual evidence codes (e.g., PVS1, PM2, PP3) were not included in the table, classification was performed using an internally customized pipeline based on the ACMG framework, as described in the Methods section. We believe the summarized classifications are sufficient for the purpose of this study, but detailed evidence codes can be provided upon request..
- Tables 1 and 2 are duplicated.
è Thank you for pointing this out. The duplication of Tables 1 and 2 has been reviewed and corrected in the revised manuscript.
- Data presentation should be sufficiently clear to allow correlation between clinical findings, biochemistry, and genetics. As the information is currently presented, such correlations cannot be made.
è Thank you for your valuable comment. We have revised Table 1 to include hormone levels presented as median values with interquartile ranges, and we expanded the Results section to describe the clinical characteristics and biochemical profiles of the patients in greater detail. These changes aim to improve the clarity of the data presentation and allow for better correlation between clinical findings, biochemical results, and genetic variants.
- As presented, data on patients with thyroid dysgenesis are misleading. Genetic testing revealed a VUS variant in Pax8 in patient #20 with thyroid dysgenesis.
è Thank you for your insightful comment. We agree that clarification is needed regarding the patients with thyroid dysgenesis.In our study, the same VUS variant in PAX8 was identified in two patients: #19 and #20. Among them, patient #19 also harbored a pathogenic variant in TSHR (p.Arg450His) and had normal findings on both thyroid ultrasonography and scintigraphy.
In contrast, patient #20, who carried only the PAX8 variant, was interpreted by a radiologist as having a subtly small thyroid gland on ultrasound. Although precise gland measurements were not available, the imaging impression was consistent with thyroid hypoplasia, and the patient was categorized accordingly in Table 1.
We have revised the Result section to reflect these distinctions more clearly. The revised text now reads:
“TSHR variants were found in three patients (9.4%).The recurrent p.Arg450His variant, classified as pathogenic, was found in two individuals (#17 and #19), both of whom had normal thyroid ultrasonography and scintigraphy findings. An additional variant, p.Ala471Thr, was identified in patient #18 and classified as a VUS. Notably, this patient has continued levothyroxine therapy (0.8 µg/kg/day) into adulthood, suggesting a persistent hypothyroidism despite the uncertain pathogenicity of the variant.
PAX8 variants were identified in two patients (6.3%), both of whom carried the same splice-site variant, c.898+1G>C, classified as VUS. Patient #20 was categorized as having thyroid hypoplasia, based on ultrasound findings of a subtly small thyroid gland, while patient #19 had normal thyroid imaging and concurrently carried the pathogenic TSHR variant (p.Arg450His).The clinical relevance of the PAX8 variant remains unclear and warrants further investigation.”

Round 2
Reviewer 2 Report
Comments and Suggestions for Authors
My comments were meant to improve the revised version of the manuscript.
- The human reference genome was not updated.
- Update allele frequency data to gnomAD v4.1.
- Does gnomAD v4.1 represent the Korean population? Database KOVA 2 (https://www.kobic.re.kr/kova/) should also be considered.
- The lack of CNV analysis should be stated in the text.
- The number of patients with GIS should be indicated in Table 1.
- Table 1 mentions patients with Prematurity and Small for Gestational Age, but no further comments are available.
- The lack of availability of the perchlorate discharge test should be stated in the text.
- The lack of family segregation analysis to confirm the trans configuration of biallelic variants should be acknowledged as a limitation of the study.
- Criteria and strength of evidence for the ACMG classification should be included in Table 2.
Author Response
Dear reviewers,
I have revised the manuscript as requested in the comments.
The answers are highlighted in yellow.
My comments were meant to improve the revised version of the manuscript.
- The human reference genome was not updated.
è Thank you for your comments. We would like to clarify that the bioinformatics analysis was indeed conducted using the GRCh38 reference genome and allele frequency data from gnomAD v4.1. However, the version labels in the initial report provided to us were not updated accordingly, which caused the discrepancy. We have now ensured that the manuscript reflects the actual versions used during the analysis. No changes to the analysis or results were necessary.
- Update allele frequency data to gnomAD v4.1.
è Thank you for your comment. We would like to clarify that the bioinformatics analysis was indeed conducted using the GRCh38 reference genome and allele frequency data from gnomAD v4.1. However, the version labels in the initial report provided to us were not updated accordingly, which caused the discrepancy. We have now ensured that the manuscript reflects the actual versions used during the analysis. No changes to the analysis or results were necessary.
- Does gnomAD v4.1 represent the Korean population? Database KOVA 2 (https://www.kobic.re.kr/kova/) should also be considered.
è Thank you for this important suggestion. While gnomAD v4.1 does not include Korean-specific allele frequencies, it provides extensive and updated data for East Asian population, which was utilized for variant interpretation in our study. Although we considered incorporating the Korean Variant Archive (KOVA2), many of the rare variants detected were absent from KOVA2. Therefore, we relied on gnomAD v4.1 as the primary source of allele frequency data to ensure analytical consistency and broader variant coverage.
- The lack of CNV analysis should be stated in the text.
è Thank you for your comment. We have now clarified in the Methods section that copy number variant (CNV) analysis was not performed in this study.
“Paired-end sequencing (150 bp × 2) was conducted using an Illumina NovaSeq 6000 platform to achieve sufficient sequencing depth for robust variant identification. Copy number variant (CNV) analysis was not performed as part of this study.”
- The number of patients with GIS should be indicated in Table 1.
è Thank you for your suggestion. We have added the number of patients with gland-in-situ (n=25) to Table 1, and updated the table accordingly.
- Table 1 mentions patients with Prematurity and Small for Gestational Age, but no further comments are available.
è Thank you for your helpful comment. We have now incorporated additional descriptions regarding the patients with prematurity and SGA status in the Results section.
“Additionally, among the study subjects, three patients were born prematurely: patients #9 and #10, who were dizygotic twins born at 27 weeks of gestation, and patient #2, born at 35 weeks. Despite their prematurity, all three exhibited normal growth and neurodevelopmental outcomes. Four patients were classified as small for gestational age (SGA) at birth. Among them, two patients — one WES-positive (patient #3) and one WES-negative — exhibited persistent short stature below the 3rd percentile and initiated growth hormone therapy. The remaining two SGA patients, including patient #6, demonstrated catch-up growth, maintaining heights around the 50th percentile. All four SGA patients also showed normal neurodevelopmental outcomes. These findings suggest that prematurity and SGA status alone may not fully account for the clinical and genetic heterogeneity observed in congenital hypothyroidism.”
- The lack of availability of the perchlorate discharge test should be stated in the text.
èThank you for your question. The perchlorate discharge test was not performed in our study, and as suggested, we have added this information to the Methods section of the revised manuscript.
“Thyroid morphology and glandular location were assessed in all participants using ultrasonography and thyroid scintigraphy. The perchlorate discharge test, which can assess defects in iodine organification, was not performed in this study because this diagnostic modality is not routinely available at our institutions. Patients were excluded if~”
- The lack of family segregation analysis to confirm the trans configuration of biallelic variants should be acknowledged as a limitation of the study.
è Continued advancements in genomic technologies, coupled with integrative functional analyses, will be essential to further elucidate the pathogenesis of congenital hypothyroidism and to facilitate the development of precision medicine approaches that improve patient outcomes.
“Although our study achieved a high mutation detection rate, several important limitations must be acknowledged. A substantial proportion of the detected variants were heterozygous findings in genes typically associated with autosomal recessive inheritance, making the direct pathogenic role of these variants uncertain. Furthermore, a heterozygous splice-site variant in PAX8, a gene generally associated with autosomal dominant inheritance, was classified as a variant of uncertain significance (VUS), and its clinical relevance remains unclear. Family segregation analysis was not performed, and thus, the trans configuration of biallelic variants could not be confirmed. In addition, functional validation studies were not conducted, limiting the interpretation of novel or uncertain variants. These limitations underscore the need for parental testing, functional assays, and larger cohort studies in future research to better define the pathogenicity and inheritance patterns of the identified variants. Despite these limitations, our findings highlight the substantial genetic heterogeneity underlying permanent congenital hypothyroidism in Korean pediatric patients and emphasize the critical role of comprehensive genetic evaluation in understanding disease etiology and clinical variability. Continued advancements in genomic technologies, coupled with integrative functional analyses, will be essential to further elucidate the pathogenesis of congenital hypothyroidism and to facilitate the development of precision medicine approaches that improve patient outcomes.”
- Criteria and strength of evidence for the ACMG classification should be included in Table 2.
è Thank you for your helpful comment. As requested, we have included the ACMG classification evidence codes (e.g., PVS1, PM2, PP3) for each variant in a newly added Supplementary Table1. This table was separated from the main results (Table 2) to improve clarity and prevent overcrowdin

Round 3
Reviewer 2 Report
Comments and Suggestions for Authors
Data analysis based on the Korean Variant Archive (KOVA2) database should be included in the manuscript.
Author Response
Comment 3: Data analysis based on the Korean Variant Archive (KOVA2) database should be included in the manuscript.
è Thank you for this valuable suggestion. We have now cross-referenced all identified variants with the Korean Variant Archive 2 (KOVA2) database. Although many of the rare variants were not found in KOVA2, allele frequency data were incorporated into Supplementary Table 1 where available. We have also updated the Genetic Analysis section of the manuscript to reflect the inclusion of KOVA2 data alongside gnomAD v4.1 and East Asian-specific allele frequencies. This addition enhances the relevance of our findings to the Korean population and improves the population-specific interpretation of genetic variants.